# Effects of in Utero SARS-CoV-2 Exposure on Newborn Health Outcomes

Margaret H. Kyle [1] and Dani Dumitriu [1,2,*]

1 Division of Child and Adolescent Health, Department of Pediatrics, Columbia University Vagelos College of Physicians and Surgeons, NewYork-Presbyterian Morgan Stanley Children's Hospital, New York, NY 10032, USA
2 Division of Developmental Neuroscience, Department of Psychiatry, Columbia Vagelos College of Physicians and Surgeons, New York, NY 10032, USA
* Correspondence: dani.dumitriu@columbia.edu; Tel.: +1-646-774-6255

**Abstract:** The severe acute respiratory syndrome coronavirus 2 (SARS-CoV-2) has infected over 600 million people worldwide, including millions of pregnant women. While newborns exposed to other viruses in utero are sometimes at high risk for vertical transmission, a substantial body of literature since early 2020 has demonstrated that vertical transmission of SARS-CoV-2 from infected mother to neonate is rare, and that newborns who do become infected with SARS-CoV-2 generally have favorable outcomes. In this review, the authors evaluate the existing literature on vertical transmission of SARS-CoV-2 and its potential mechanisms and discuss short- and long-term health outcomes in newborns who were exposed to SARS-CoV-2 in utero. The authors conclude that vertical transmission and adverse neonatal and infant/child outcomes are unlikely, but that neonates exposed to prenatal maternal SARS-CoV-2 infection may be at slightly higher risk for preterm birth, possibly related to increased risk of severe COVID-19 disease in pregnant women, placental changes, or infection timing. Ultimately, the need for additional and longer-term follow-up data in this population is highlighted.

**Keywords:** COVID-19; SARS-CoV-2; pregnancy; newborn; neonatal; vertical transmission; placenta; neurodevelopment

## 1. Introduction

Two and a half years into the COVID-19 pandemic, the severe acute respiratory syndrome coronavirus 2 (SARS-CoV-2) continues to spread worldwide. As of this writing (12 September 2022), there have been over 600 million cases globally, with over four million new cases reported in the most recent World Health Organization report for the week of 29 August 2022 [1]. The global health burden of COVID-19 disease has affected the entire world, but certain populations, including pregnant women and newborns, have been considered particularly vulnerable to potential adverse outcomes following SARS-CoV-2 infection. One fear among researchers and clinicians early on was the possibility of vertical transmission of SARS-CoV-2 from infected pregnant women to their neonates, but a large body of literature has demonstrated that this occurs in a small proportion of neonates, generally less than 5% [2–7]. Further, rates of neonatal infection do not appear to be increased when newborns breastfeed, room-in, or partake in skin-to-skin care with their mothers [5–7]. Potential mechanisms underlying rare cases of vertical transmission remain unclear, but a prevailing theory is transplacental transmission. Additionally, SARS-CoV-2 has been linked to placental alterations with the capacity to impact newborn outcomes even in the absence of placental infection [8–10]. Available data suggest that newborns exposed to maternal SARS-CoV-2 infection in utero are generally healthy, but that they may be at increased risk of premature birth [6,11–14]. The literature on long-term outcomes in these newborns is—by definition—limited. Here, the authors review the present data on

vertical transmission of SARS-CoV-2 and its potential mechanisms, consider the role of the placenta in neonatal outcomes, and survey data on short- and long-term outcomes in neonates exposed to maternal SARS-CoV-2 infection in utero (Table S1).

## 2. Vertical Transmission of SARS-CoV-2 from Infected Mother to Newborn

### 2.1. Frequency of Vertical Transmission of SARS-CoV-2

Vertical transmission is defined as the transmission of an infection from an infected woman to her newborn, and may occur during intrauterine, intrapartum, or postnatal periods [15]. Possible vertical transmission of SARS-CoV-2 from infected mothers to their neonates was a source of significant concern for clinicians and researchers early in the COVID-19 pandemic, given the immature neonatal immune system [16–18]. For several viruses, including Hepatitis viruses and HIV, vertical transmission represents the major mechanism whereby infants and children develop viral infection [15,19,20]. Fortunately, a multitude of evidence since the beginning of the COVID-19 pandemic has indicated that vertical transmission of SARS-CoV-2 from infected mother to neonate is rare. Many studies, including national registries and large systematic reviews, have shown that the prevalence of vertical transmission in neonates exposed to SARS-CoV-2 in utero is generally around 5% or lower. A meta-analysis including 10,000 women infected with SARS-CoV-2 during pregnancy found a vertical transmission rate of 5.3% in exposed newborns [3], and another systematic review on 936 neonates born to SARS-CoV-2-infected mothers detected a vertical transmission rate of 3.2% [2]. Both of these analyses included reports from a wide range of countries and regions including Asia, Europe, and the Americas [2,3]. Another analysis co-reporting on 4005 pregnant women included in the American Academy of Pediatrics Section on Neonatal-Perinatal Medicine (AAP-SONPM) or the UK and Global Pregnancy and Neonatal outcomes in COVID-19 (PAN-COVID) Registry found that vertical transmission rates were 1.8% in the US registry and 2.0% in the UK registry [4].

Vertical transmission of viruses is generally thought to occur through transplacental transmission (see Section 2.2.1 below), cervicovaginal transmission during delivery, or contact with the mother during the postnatal period [19]. In addition to the low vertical transmission rates cited above, the available evidence further demonstrates that postnatal practices like vaginal delivery, breastfeeding, and rooming-in with mothers do not appear to put neonates at increased risk of infection. A systematic review by Walker et al. reported on 666 neonates born to SARS-CoV-2-infected women, 4% of whom tested positive for the virus [7]. The authors calculated separate neonatal infection rates depending on mode of delivery and postnatal care practices and found that 2.7% of vaginally delivered neonates tested positive as opposed to 5.3% of neonates born by cesarean section, and that 4.7% of breastfed newborns tested positive, a rate similar to the 5.3% of formula-fed newborns who tested positive [7]. Newborns who were allowed to room-in with their mothers also tested positive less frequently (3.7%) than those who were isolated (13%), although isolation may have occurred as a result of positive testing so this difference should be interpreted with caution. Similarly, our group reported on 101 newborns born to women infected with SARS-CoV-2 perinatally and found that 2.0% of exposed neonates tested positive for the virus, and that there was no clinical evidence of COVID-19 disease in any neonate despite encouraging rooming-in, breastfeeding, and skin-to-skin care [5]. A national cohort study from Sweden, where rooming-in, breastfeeding, and skin-to-skin care were also encouraged, reported on over 2000 newborns born to SARS-CoV-2-infected women and determined a vertical transmission rate of just 0.9% [6]. Likewise, a cohort of 177 newborns in Spain who all exclusively breastfed had a neonatal SARS-CoV-2 infection rate of 5.1% [21]. The systematic review by Kotlyar et al. that found a vertical transmission rate of 3.2% also separated rates between studies conducted in China and studies outside of China and determined that 2.0% of the neonates studied in China tested positive, as opposed to 3.5% of neonates studied outside of China [2]. It is reassuring that these rates are similar and both low, given that China has encouraged more stringent postnatal care policies to prevent neonatal infection, such as separating exposed neonates from their mothers

and prohibiting breastfeeding [22]. Overall, vertical transmission appears to be unlikely in neonates exposed to maternal SARS-CoV-2 infection, and postnatal care policies like rooming-in, breastfeeding, and skin-to-skin contact that have demonstrated benefits for newborns [23–25] do not seem to augment this low risk, and are possibly even protective.

*2.2. The Role of the Placenta in Vertical Transmission of SARS-CoV-2 and Neonatal Outcomes*

2.2.1. Transplacental Transmission of SARS-CoV-2

Although vertical transmission of SARS-CoV-2 from mother to neonate is demonstrably rare, it is still necessary to determine the mechanism by which it may occur in infants who do become infected. Proposed mechanisms have included cervicovaginal transmission during delivery and transplacental transmission during gestation. Researchers have tested placental tissue, amniotic fluid, cord blood, and vaginal secretions from infected mothers and their neonates and shown that viral RNA is rarely detected in any of these samples using RT-PCR. A meta-analysis including 563 neonates born to SARS-CoV-2-infected mothers showed that SARS-CoV-2 was detected in 12% of placental samples and 4–6% of amniotic fluid, cord blood, and vaginal secretions tested [3]. Similarly, the above systematic review including 936 neonates born to SARS-CoV-2-infected mothers found that 7.7% of placental samples, 0% of amniotic fluid samples, and 2.9% of cord blood samples tested positive in their analysis [2]. Infection of any of these samples appears to be rare, but, notably, both analyses found higher rates SARS-CoV-2 positivity in placental samples than other samples, suggesting that transplacental transmission is more likely than cervicovaginal transmission.

The mechanism that SARS-CoV-2 uses to infect cells further implicates transplacental transmission as a possible mechanism for rare cases of vertical transmission. SARS-CoV-2 depends on two key viral factors to enter host cells: serine protease TMPRSS2 primes viral spike protein domains, facilitating cell entry using SARS-CoV receptor angiotensin-2 converting enzyme (ACE2) [26,27]. ACE2 and TMPRSS2 are highly expressed in the respiratory and gastrointestinal tracts [28], but they have also been found in placental trophoblasts [29,30], leading to speculation that placental infection and subsequent transplacental vertical transmission of SARS-CoV-2 are possible. However, transplacental infection requires co-expression of ACE2 and TMPRSS2 in placental cells as well as maternal viremia [31]. Consistent with the aforementioned low rates of vertical transmission of SARS-CoV-2, the evidence to-date has demonstrated that this co-expression is rare in placental cells [32–34], and that viremia is uncommon in SARS-CoV-2 patients [35], particularly in pregnant women [33]. Several groups have studied placental samples from SARS-CoV-2-infected mothers and found no evidence of placental infection and no difference in placental histopathology between samples taken from infected patients and those taken from uninfected patients [33,36,37]. Interestingly, even studies that found placental infection in tissue samples failed to show evidence that it predicts actual neonatal infection and vertical transmission [38–41].

To further probe the possible role of the placenta in vertical transmission, a recent study examined co-expression of ACE2 and TMPRSS2 in the placenta and in a variety of fetal tissue samples [34]. The authors found that the placenta as well as most fetal tissues analyzed did not co-express ACE2 and TMPRSS2. Only the fetal gastrointestinal tract co-expressed both proteins, suggesting that fetal infection would require swallowing of infected amniotic fluid [34]. However, in further support of the infrequency of vertical transmission, amniotic fluid is typically only infected in cases of viremia and severe disease [42].

2.2.2. Other Placental Mechanisms Related to Maternal SARS-CoV-2 Infection

The existing evidence suggests that, despite being the most plausible mechanism of vertical transmission of SARS-CoV-2, transplacental transmission from infected mother to neonate is rare and unlikely given the cellular framework required for infection. Nonetheless, placental tissue remains an important area of study in SARS-CoV-2-infected women,

and recent findings indicate that adverse placental changes with the potential to impact neonatal outcomes may occur in the absence of placental infection or vertical transmission. Early in the pandemic, a research group in Chicago examined placental histopathology in 16 patients with SARS-CoV-2 during pregnancy and found that maternal vascular malperfusion (MVM) was more common in infected women compared to historical controls [9]. MVM is a pattern of placental injury related to altered blood flow to the placenta [43], and it has been linked to increased rates of adverse maternal and neonatal outcomes including preeclampsia, fetal death, small for gestational age, and preterm delivery [44]. The same group published a recent study on placental lesions in 883 women who were infected with SARS-CoV-2 during pregnancy and 185 controls with no history of infection during pregnancy or vaccination against COVID-19 [8]. Similarly, they found that MVM was significantly more common in the infected group, with 63% of women infected during pregnancy exhibiting MVM features as opposed to 47% of uninfected women. Interestingly, the authors found that MVM frequency varied depending on the stage of the pandemic during which the infection occurred, with 82% of women infected in the Delta variant era exhibiting MVM, 65% in the Alpha/Gamma era and 55% in the Omicron era [8], suggesting that viral variants may have differing effects on placental function. All Omicron patients included in the study were infected during the third trimester of pregnancy, though, so it is also possible that infection timing during gestation impacts the frequency of MVM in pregnant patients infected with SARS-CoV-2. In contrast, two other studies with sample sizes of 65 and 101 SARS-CoV-2 infected women [36,37] have shown no differences in placental pathologic features, including MVM, between infected and uninfected mothers. Of note, all studies reporting on placental lesions included in this review used the Amsterdam Criteria [45] for placental examination, so these differing findings do not appear to be related to different standardization criteria.

Given that pregnant women seem to be more susceptible to severe COVID-19 disease (see Section 3.2.2), it is also imperative to determine whether placental pathology correlates with disease severity, potentially predisposing severely ill women and their neonates to adverse pregnancy and birth outcomes. Mourad et al. [10] studied placental samples from 66 women infected with SARS-CoV-2 late in pregnancy and found that disease severity did not predict placental histopathological features, but that placental expression of ACE2 and interferon-induced transmembrane antiviral genes was higher in severely ill women, suggesting that severe disease alters the placental response to SARS-CoV-2. Contrasting this finding, the above discussed study that found increased frequency of MVM in infected women [8] also showed that MVM frequency increased with disease severity. Overall, the existing evidence regarding effects of maternal SARS-CoV-2 infection on the placenta is limited and inconclusive, but given several studies showing an altered placental environment and, separately, increased risk of preterm delivery in pregnant women with COVID-19 disease (see Section 3.2.1), it is essential that the field continues to disentangle this relationship.

## 3. Clinical Outcomes in Newborns Exposed to SARS-CoV-2 In Utero

### 3.1. Outcomes in Infected Newborns

Although vertical transmission of SARS-CoV-2 from mother to newborn is rare, outcomes must be followed in the small proportion of newborns who do become infected perinatally. The newborn immune system is immature, and newborns are especially susceptible to respiratory infections [16–18], making them a population potentially vulnerable to severe COVID-19 disease. The current literature is conflicting, but some reports have suggested increased risk in newborns infected with SARS-CoV-2. A population-level report on neonatal infection early in the pandemic in the UK found that the rate of infection in newborns was low, at 0.05–0.06% of the newborn population, but that those who were infected required intensive care or respiratory support at a high rate of 36% [46]. However, this figure is confounded by the fact that 24% of the infected infants had been born preterm and may have required respiratory support for reasons unrelated to SARS-CoV-2

infection. A study from China found that infants under one year old had severe COVID-19 disease at a higher rate than other pediatric populations, but this proportion was still low, with 10.7% of infants classified as having severe or critical disease [47], as compared to 19% of cases in the overall Chinese population classified as severe or critical during the same period [48].

In contrast to initial reports of severe disease, several studies have described asymptomatic or mild COVID-19 disease courses in newborns [6,49–52]. Early in the pandemic, two reports from large medical centers in New York [50] and Chicago [49], respectively, found that none of the newborns seen in either hospital system who had COVID-19 disease required respiratory support. Another report from New York found that 18 of 20 symptomatic newborns infected with SARS-CoV-2 and seen in the study's hospital system from March through April 2020 had a mild disease course [52]. The remaining two newborns required supplemental oxygen, but did not require intubation or other intensive intervention [52]. The national cohort from Sweden described above found that none of the 21 infants who tested positive for SARS-CoV-2 during the neonatal period had any morbidity related to COVID-19 disease [6]. Similarly, a study in India that followed outcomes in neonates with SARS-CoV-2 infection showed that a majority of the 21 neonates were asymptomatic, with only seven being diagnosed with symptomatic COVID-19 disease [53]. Three neonates required mechanical ventilation (two non-invasive, one invasive), but these neonates were born between 30- and 33-weeks gestation, and all term infants with symptomatic COVID-19 disease required only supportive care. Follow-up of 20 of the neonates two months after discharge found that none had exhibited any further symptoms or required repeat hospitalization [53]. A recent systematic review including 1214 children under the age of five, half of whom were infants under one year old, found that over 90% of children had mild or moderate disease, while only 7% had severe disease and one child died [54]. Of note, most of the available literature on newborn outcomes has come from early in the pandemic, when the Alpha/Gamma strains were dominant. More data is needed to determine differences in risk in newborns infected with SARS-CoV-2 during later variants, especially given the increased transmissibility of the recent Omicron variant and associated increased rates of hospitalization in infants and children [55]. One recent study from Massachusetts has shown that infants and children infected during the Omicron wave of the pandemic were more likely to be diagnosed with COVID-19 associated croup, but that, as has frequently been reported with earlier viral variants, no child required invasive ventilation [56]. While more studies on outcomes in newborns infected with SARS-CoV-2 are needed, the existing data indicate that newborns seem to be at lower risk of severe COVID-19 disease and death than the general population, and that adverse outcomes in neonates can typically be attributed to prematurity or other causes unrelated to COVID-19 disease.

### 3.2. Neonatal Outcomes in Uninfected Newborns Exposed to SARS-CoV-2 In Utero
#### 3.2.1. Increased Risk of Preterm Birth in Newborns Exposed to SARS-CoV-2 In Utero

Newborns who were exposed to SARS-CoV-2 in utero but do not become infected still could be at risk for adverse outcomes, considering the demonstrated risk of delayed growth, congenital anomalies, and preterm delivery (i.e., delivery at less than 37 weeks gestation), in pregnant women with other infections and/or elevated inflammation [57–59]. The current literature indicates that maternal SARS-CoV-2 infection during pregnancy may indeed increase the risk of preterm delivery: an early study from Spain on 503 newborns exposed to SARS-CoV-2 in utero found a preterm birth rate of 15.7%, more than double the national average in Spain of 7.5% [11]. Likewise, a co-report on national registries from the US and UK including over 4000 women infected with SARS-CoV-2 during pregnancy between January and July of 2020 [4] found a 15.7% preterm birth rate in the US registry and a 12.0% preterm birth rate in the UK registry, both of which are higher than the national average rates of 10% in the US [60] and 7.5% in the UK [61]. The report from Sweden on over 2000 newborns born to mothers infected with SARS-CoV-2 during pregnancy found that

these newborns appeared to have higher rates of neonatal respiratory disorders compared to case-matched controls, but this increase was mediated by their increased prevalence of preterm birth [6]. A study on 734 women in Brazil similarly found that women infected with SARS-CoV-2, and specifically those with moderate and severe COVID-19 disease, were more likely to give birth preterm and to have neonates admitted to intensive care [13]. An 18 country, multinational cohort study found further evidence of increased risk of preterm delivery in 706 SARS-CoV-2-infected women who gave birth in 2020 as compared to 1424 uninfected women, with women with symptomatic COVID-19 disease delivering on average 0.8 weeks earlier than uninfected women [14]. More recent data collected during the Omicron variant era in New York support these findings, showing a significantly increased risk of preterm birth in 631 women who were infected with SARS-CoV-2 between December 2021 and February 2022, as compared to 4107 uninfected women [12]. Conversely, two recently published studies that used data from the Alpha/Gamma variant eras found no difference in rates of preterm birth between SARS-CoV-2-infected and uninfected pregnant women in a cohort of 116 infected women in New York [62], and a six-country cohort of 510 infected women in sub-Saharan Africa [63]. While interesting, these reports had smaller sample sizes than many of the studies from the same time period that found increased risk of preterm birth, including multiple national registries. Studies that have reported on other neonatal outcomes, including Apgar scores, congenital anomalies, and small for gestational age have found no indication of abnormalities in neonates born to SARS-CoV-2-infected mothers [4,6,11,14,62,63]. Considering the evidence together, it appears that newborns who were exposed to SARS-CoV-2 in utero are at an increased risk of premature birth, but differences in gestational age at delivery between exposed and unexposed newborns are overall small, at less than one week [14]. Encouragingly, none of these studies have indicated that newborns born to SARS-CoV-2-infected mothers show increased rates of any morbidity beyond the morbidity associated with prematurity itself.

### 3.2.2. Possible Mechanisms Driving Increased Risk of Preterm Birth

The cause of increased rates of preterm delivery in women with SARS-CoV-2 infection during pregnancy remains unclear. One hypothesis is that increased risk of preterm delivery is related to the increased prevalence of severe COVID-19 disease in pregnant women. This is supported by several large studies and national registries that have shown that pregnant women are at higher risk of severe COVID-19 disease and intensive care unit admission. A national registry from Mexico comparing 5183 pregnant women with SARS-CoV-2 infection to 5183 non-pregnant women with SARS-CoV-2 infection and closely matched on demographic variables and underlying risk factors such as smoking, asthma, obesity, and cardiovascular disease found that pregnant women were at a significantly higher risk for death, pneumonia, and intensive care unit admission [64]. Similarly, an analysis from the Centers for Disease Control and Prevention on 400,000 women found that pregnant women with symptomatic COVID-19 disease were at higher risk than non-pregnant women with symptomatic COVID-19 disease for receiving invasive ventilation, receiving extracorporeal membrane oxygenation, being admitted to an intensive care unit, and death [65]. This analysis also adjusted for age, race, ethnicity, and underlying conditions. The most recent version of a large, living (i.e., continuously updated) systematic review on outcomes in pregnant women with SARS-CoV-2 infection included 435 studies, over 100,000 pregnant or recently pregnant women with COVID-19 disease, and 2 million non-pregnant women of reproductive age with COVID-19 disease [66]. The authors found that pregnant women were more likely to be admitted to the intensive care unit or need invasive ventilation, and were less likely to exhibit mild to moderate symptoms like fever, cough, dyspnea, and myalgia [66].

Consistent with this hypothesis that increased rates of preterm birth in women with SARS-CoV-2 infection may be driven by increased disease severity in this population, recent studies have found that disease severity predicts preterm delivery. A study pooling data on 1219 women with SARS-CoV-2 infection who delivered at 33 US hospitals showed

that severe or critical COVID-19 disease was associated with higher risk of preterm birth compared to asymptomatic COVID-19 cases, but that there was no difference in risk between asymptomatic cases and mild to moderate cases [67]. Similarly, a study on over 6000 pregnant women with SARS-CoV-2 in the US found that critical COVID-19 disease was associated with increased risk of preterm birth compared to mild disease [68]. The Brazilian study mentioned above further supports this idea, finding that women with moderate and severe COVID-19 disease, but not those with asymptomatic or mild disease, were more likely than women with no SARS-CoV-2 infection during pregnancy to deliver preterm [13]. Studies on other respiratory diseases have also shown that increased disease severity is associated with preterm delivery [69], and this relationship is thought to be driven by the medical interventions and drug treatments necessary in cases of severe illness [69]. Placental infection related to other viruses has also been associated with preterm birth [70], and, given some reports of changes in the placenta in SARS-CoV-2-infected pregnant women reviewed in Section 2.2.2, particularly in cases of severe disease, placental alterations may present a possible mechanism for the increased rates of preterm birth seen in women with COVID-19 disease.

Another emerging hypothesis is that the timing of maternal SARS-CoV-2 infection relative to gestation may predict adverse neonatal outcomes like preterm delivery. It is well-established that prenatal insults affect fetal development at insult-specific critical periods during pregnancy [71], and studies on maternal malaria and influenza infection have shown that increased rates of preterm birth and low birth weight in exposed infants are driven by first or second trimester maternal infection [72–74]. Accordingly, a recent study comparing outcomes between 19,769 SARS-CoV-2-uninfected women and 882 SARS-CoV-2-infected women separated into trimester of infection groups (85 with first trimester infection, 226 with second trimester infection, and 571 with third trimester infection) found that increases in risk of preterm delivery and stillbirth in the infected group were driven by first and second trimester infections [75]. Further, the authors showed that gestational age at the time of infection was correlated with gestational age at delivery, and that maternal disease severity did not predict gestational age at delivery [75]. Another study including 402 pregnant women with SARS-CoV-2 infection in the first or second trimester and 11,705 uninfected women likewise found that infection in the first or second trimester was associated with increased risk of preterm birth, but not increased risk of moderately to extremely preterm birth (here defined as less than 34 weeks gestation), and with a slightly increased risk of fetal or neonatal death [76]. In contrast, a larger retrospective cohort study from Israel including 2753 SARS-CoV-2-infected pregnant women (478 with first trimester infection, 943 with second trimester infection, and 1332 with third trimester infection) found that third trimester infection, but not first or second trimester infection, was associated with increased risk of preterm birth [77]. Another study followed outcomes in 16 first trimester SARS-CoV-2-infected women and found no difference compared to uninfected women [78]. Clearly, the evidence on SARS-CoV-2 infection timing during pregnancy and risk of preterm birth is limited and conflicting, but, given our knowledge of trimester-specific effects of other viruses and environmental insults, this is an area that should be further explored.

## 4. Long-Term Outcomes in Infants and Children Exposed to SARS-CoV-2 In Utero

While immediate neonatal outcomes are largely favorable in newborns exposed to SARS-CoV-2 in utero, there is a need for longer-term research into their child development. Studies on in utero HIV-exposed uninfected children have found that exposure is associated with higher risk for neurodevelopmental delay [79,80]. The intrauterine inflammation that may result from maternal viral infection (with any virus) has been linked to epilepsy, autism spectrum disorder, and schizophrenia in offspring [81]. To investigate early indicators of this risk, our group compared Ages and Stages Questionnaire, Third Edition (ASQ-3) scores at six months of age between 114 infants exposed to SARS-CoV-2 in utero and 141 infants not exposed, and found no significant differences on any subdomain score [82]. Interestingly, though, our group found that infants born during the pandemic,

irrespective of maternal infection status, had lower ASQ-3 scores in gross motor, fine motor, and personal-social domains when compared to a historical control group born prior to the pandemic, presumably mediated through maternal stress during pregnancy [82]. Another report from our group used the Infant Behavioral Questionnaire when infants were six months old and also found no differences in temperament between infants exposed vs. unexposed to SARS-CoV-2 in utero, but an association between maternal postnatal COVID-19 pandemic-related stress and both an increase in infant negative temperament and a decrease in infant positive temperament [83]. A study from Kuwait also used the ASQ-3 at 10–12 months of age in 298 infants exposed in utero to maternal SARS-CoV-2 infection, and found that 10% of infants assessed had a score that was at least two standard deviations below the population mean, indicating a possible developmental delay [84]. This report did not have a control group, but the 10% rate of developmental delay was lower than a study from Lebanon–cited by the authors as having a similar geographical and cultural setting as Kuwait–that found a pre-pandemic rate of developmental delay of 15% in healthy infants using the ASQ-3 [85], underscoring our group's findings that in utero SARS-CoV-2 exposure does not appear to predict early neurodevelopmental deficits. An ongoing longitudinal study from Italy that has followed auxologic and neurologic outcomes in infants exposed to SARS-CoV-2 in utero after discharge has also found no abnormal findings in any exposed infants during follow-up at 3, 6, and 9 months of age [86].

In contrast to the accumulating evidence against an association between in utero exposure to maternal SARS-CoV-2 infection and adverse neurodevelopmental effects, a recent study from Boston reported that 222 infants exposed to maternal SARS-CoV-2 infection in utero were more likely than 7550 unexposed infants to have received a neurodevelopmental diagnosis in the first year of life using the International Classification of Diseases, Tenth Revision (ICD-10) [87]. However, this study suffered from several limitations. First, it should be noted that only 14 of 222 (6.3%) infants in the exposed group had received a neurodevelopmental diagnosis, compared to 227 of 7550 (3.0%) in the unexposed group [87], so the rates in both groups were low. The study used retrospective chart review and therefore was limited in its control over confounding variables, and one year of age is generally considered to be too early to diagnose most neurodevelopmental disorders, including autism spectrum disorder [88]. Importantly, the study also suffered from the additional confounder of limited SARS-CoV-2 testing in Boston during the study period, reporting a positivity rate of only 2.9% in the study sample, during a period of time when the test positivity rate was 30–40% on general testing in the Boston area [89] and 15.4% on universal testing of delivering patients in New York City [90], a city with similar pandemic course. The increase in ICD-10 codes was also not statistically significant when full-term infants were evaluated separately, suggesting that prematurity mediated their main observed effect. Finally, the lower median age when infants received their ICD-10 code in the exposed group suggests increased parental concern as a strong mediating factor. Considering these methodological concerns along with the four studies discussed above that have demonstrated no differences in developmental outcomes between in utero SARS-CoV-2 exposed and unexposed infants, the currently available data on long-term outcomes and neurodevelopment in infants exposed to SARS-CoV-2 in utero overall suggests that these infants do not seem to be at any significantly increased risk of neurodevelopmental delay. However, more studies and longer-term follow-up are necessary to conclusively determine risk in this population of infants and children. In particular, there is a paucity of data on neurodevelopmental effects in infants born to mothers with a history of severe or critical COVID-19 disease, as well as a paucity of data from women infected early in pregnancy.

## 5. Conclusions

In sum, the literature on neonates exposed to maternal SARS-CoV-2 infection in utero indicates that these newborns generally have favorable outcomes. Vertical transmission is rare, even when newborns are roomed-in with mothers, breastfeed, and practice skin-

to-skin care. In accordance with the rarity of vertical transmission of SARS-CoV-2, the mechanism by which vertical transmission may occur remains unclear: placental infection appears to be the most plausible mechanism, but it has been reported infrequently. In the rare cases that neonates do become infected with SARS-CoV-2, they tend to fare well, with very few exhibiting severe disease or requiring intensive care. Regardless of their neonatal infection status, newborns born to SARS-CoV-2-infected women do seem to have a slightly increased risk of preterm birth, which may be related to increased risk of severe COVID-19 disease in pregnant women, alterations in the placental environment, trimester of infection, or a combination of these and other factors. However, these newborns do not appear to be at risk of extreme prematurity or any other adverse neonatal outcome. Further, the limited long-term follow-up data available on these newborns is promising and indicates that their development to-date is normal, but, notably, outcomes for these infants beyond age 12 months are not yet available. Many of the child health outcomes previously associated with maternal infection and inflammation during pregnancy, including autism spectrum disorder, schizophrenia, and asthma, do not emerge until later in childhood. With this in mind, there is a strong need for more studies on development in infants and children exposed to SARS-CoV-2 in utero in order to determine long-term outcomes in this population.

- Newborns exposed to maternal SARS-CoV-2 infection in utero rarely acquire SARS-CoV-2 infection through vertical transmission, even when rooming-in, breastfeeding, and skin-to-skin care are practiced.
- The mechanism underlying rare cases of vertical transmission is unclear: placental infection is the most plausible mechanism but it occurs infrequently.
- Infants infected with SARS-CoV-2 (whether vertically or through community transmission) tend to have favorable outcomes, and very few infants require intensive care or exhibit severe disease.
- Newborns born to SARS-CoV-2 infected women have a slight increased risk of preterm birth, with possible underlying mechanisms including but not limited to severe COVID-19 disease in pregnant women, placental alterations, or trimester of maternal infection.
- Limited long-term follow-up data on infants and children exposed to SARS-CoV-2 in utero suggest that their development is normal, but there is a strong need for longer-term follow-up in these children.

**Supplementary Materials:** The following supporting information can be downloaded at: https://www.mdpi.com/article/10.3390/encyclopedia3010002/s1, Table S1: Summary of Literature.

**Funding:** This research was supported by National Institutes of Health R01MH126531 to D.D., Centers for Disease Control contract 75D30120C08150 through Abt Associates to D.D., and gift funds from Einhorn Collaborative to the Nurture Science Program.

**Acknowledgments:** The authors would like to thank the COVID-19 Mother Baby Outcomes (COMBO) Initiative team at Columbia University Irving Medical Center (www.ps.columbia.edu/COMBO) (accessed on 22 December 2022), and the Maternal-Child Research Oversight (MaCRO) Committee and the Departments of Pediatrics, Psychiatry, and Obstetrics and Gynecology at Columbia University Irving Medical Center for their institutional support.

**Conflicts of Interest:** D.D. has pending consultation fees from Medela for work unrelated to this review.

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
