# Peer review of "Effects of in Utero SARS-CoV-2 Exposure on Newborn Health Outcomes"

_encyclopedia, doi:10.3390/encyclopedia3010002_

Round 1
Reviewer 1 Report
Dear Authors,
This is an interesting topic, however I found some issues that you should considfer:
1. It is mentioned in the introduction that vertical transmission of SARS-CoV-2 from infected mother to neonate is rare. If it is rare, please give a sufficient explanation why is the manuscript is important? I guess it is likely less important in terms of disease transmission to other individuals. Please explore in more details in what aspect the study in the manuscript may regarded as significantly important. (lines 51 and 65).
2. Is there any logical explanation regarding the fact you found from literature that newborns who were allowed to room-in with their mothers were tested positive less frequently (line 72-73). It seems counter-intuitive.
3. There should be sufficient explanation regarding several type of vertical transmission, such as physical contact, transplacental transmission etc.
4. What is the difference between vertical transmission and perinatal transmission. Are they different or the same in terms of disease transmission.
5What is exactly the firm definition of vertical transmission, and how the exact mechanism of vertical transmission? I suggest the authors give a more strong definition.
Author Response
Reviewer 1:
This is an interesting topic, however I found some issues that you should consider:
1. It is mentioned in the introduction that vertical transmission of SARS-CoV-2 from infected mother to neonate is rare. If it is rare, please give a sufficient explanation why is the manuscript is important? I guess it is likely less important in terms of disease transmission to other individuals. Please explore in more details in what aspect the study in the manuscript may regarded as significantly important. (lines 51 and 65).
We added lines 53-59 to explain the importance of studying vertical transmission: primarily, newborns have an immature immune system that may be more vulnerable to respiratory viruses, and for several other viruses, vertical transmission is the major mechanism whereby newborns become virally infected. Thus, it is critical to consider whether or not this is possible with SARS-CoV-2, and to consider mechanisms by which vertical transmission may occur in SARS-CoV-2.
- Is there any logical explanation regarding the fact you found from literature that newborns who were allowed to room-in with their mothers weretested positive less frequently (line 72-73). It seems counter-intuitive.
In our discussion of this finding we explain that this finding may have been a result of some isolated newborns being isolated because they had tested positive: “Newborns who were allowed to room-in with their mothers also tested positive less frequently (3.7%) than those who were isolated (13%), although isolation may have occurred as a result of positive testing so this difference should be interpreted with caution.”
There should be sufficient explanation regarding several type of vertical transmission, such as physical contact, transplacental transmission etc.
We clarified in the text that the section from lines 74-105 is focused on the postnatal contact form of vertical transmission, whereas Section 2.2.1 discusses the cervicovaginal and transplacental routes. These are the three proposed mechanisms of viral vertical transmission.
- What is the difference between vertical transmission and perinatal transmission. Are they different or the same in terms of disease transmission.
Perinatal/postnatal transmission is a subset of vertical transmission that occurs after delivery – we clarified this wording in the manuscript.
5What is exactly the firm definition of vertical transmission, and how the exact mechanism of vertical transmission? I suggest the authors give a more strong definition.
We added a definition to lines 53-55 and an overview of mechanisms to lines 74-76.
Reviewer 2 Report
This manuscript entitled 'Effects of in utero SARS-CoV-2 exposure on newborn health outcomes' is a very well written and easy understandable review, absolutely worth publication.
But, the authors may consider to fill in illustrations sorting the different studies and make the results more visible.
E.g. how many placental studies with/without traces of infection? What kind of morphology? Coincidentally villitis? How many publications with MVM?
Recommend to implemate short comment on how standardization was used for placental diagnosis? etc...
Thanks to the authors!
Author Response
Reviewer 2:
This manuscript entitled 'Effects of in utero SARS-CoV-2 exposure on newborn health outcomes' is a very well written and easy understandable review, absolutely worth publication.
But, the authors may consider to fill in illustrations sorting the different studies and make the results more visible.
E.g. how many placental studies with/without traces of infection? What kind of morphology? Coincidentally villitis? How many publications with MVM?
We added Table 1 to summarize the literature, to make it easier for readers to visually see the number of studies discussed with each outcome and the sample sizes of each of those studies. We also added the data collection period to contextualize each study with its timing during the pandemic.
Recommend to implemate short comment on how standardization was used for placental diagnosis? etc... Thanks to the authors!
We added the following to lines 177-179 on placental diagnostic criteria: “Of note, all studies reporting on placental lesions included in this review used the Amsterdam Criteria (45) for placental examination, so these differing findings do not appear to be related to different standardization criteria.”
Reviewer 3 Report
Q:Line 96-134:
The authors provide several pieces of evidence to support that placenta may be the target for the COVID-19. However, which subtype or variant of the COVID-19 has a highest infection rate in the placenta? (to echo with your line 156-157)
Line 72-78 Room-in
I appreciate your findings, and your statement in these sentences. However, did you consider that the structure of the ward in the hospital is different in different countries, for example like USA and the Republic of The Gambia. It seems that the references you chose in this setion, such as Sweden, USA, etc (China is an exceptional case, because it so far still taking a very stringent law in preventing the pandemics).The design and the structure of the ward may enhance of decrease the pandemic spread. Women chose to room-in in different countries cannot be comparable, unless these factors are considered.
Line 355-359: the maternal stress is specific to the COVID-19. Dose this mean that the stress from the infection of COVID-19 is different from the stress from other diseases leading to a negative impact on the infant temperament??
Line 393-396: The current data? Dose this sentence reflected the previous section (line 371-391)? If so, may I ask the authors how these previous diagnosed “developmental delay” due to maternal exposed COVID-19 become “normal”? Can you add extra comments to the good news? (due to a good response to the rehabilitation, or the inflammation effects automatically attenuation, or the inflammation may not be persist so long as to affect their on-going neurological development?)
Author Response
Reviewer 3:
Q:Line 96-134:The authors provide several pieces of evidence to support that placenta may be the target for the COVID-19. However, which subtype or variant of the COVID-19 has a highest infection rate in the placenta? (to echo with your line 156-157)
To our knowledge, no group has reported on placental infection rates between COVID-19 variants; the only paper we have identified that reported on placental differences by COVID-19 variant/era (Shanes et al.; citation 8) reported on placental histopathology, not infection.
Line 72-78 Room-in: I appreciate your findings, and your statement in these sentences. However, did you consider that the structure of the ward in the hospital is different in different countries, for example like USA and the Republic of The Gambia. It seems that the references you chose in this setion, such as Sweden, USA, etc (China is an exceptional case, because it so far still taking a very stringent law in preventing the pandemics).The design and the structure of the ward may enhance of decrease the pandemic spread. Women chose to room-in in different countries cannot be comparable, unless these factors are considered.
We agree that the hospital setting is an important consideration when considering vertical transmission and viral spread. The sources that we used to discuss rates of vertical transmission came from a wide range of regions, including three systematic reviews that included all worldwide reports on vertical transmission at the time of review. Our aim in specifically mentioning countries in which rooming-in and breastfeeding were encouraged (e.g. the USA, Sweden) was to demonstrate that rates of vertical transmission actually remain similar, compared to worldwide systematic reviews, despite the large differences in hospital policies and ward structures. We explain in lines 99-105: “It is reassuring that these rates are similar and both low, given that China has encouraged more stringent postnatal care policies to prevent neonatal infection, such as separating exposed neonates from their mothers and prohibiting breastfeeding (22). Overall, vertical transmission appears to be unlikely in neonates exposed to maternal SARS-CoV-2 in-fection, and postnatal care policies like rooming-in, breastfeeding, and skin-to-skin contact that have demonstrated benefits for newborns (23-25) do not seem to augment this low risk”
We further demonstrate in Table 1 that the rates of vertical transmission cited from our sources are quite similar and all around or under 5%, despite the large spread of regions and associated hospital policies included.
Line 355-359: the maternal stress is specific to the COVID-19. Does this mean that the stress from the infection of COVID-19 is different from the stress from other diseases leading to a negative impact on the infant temperament??
The COVID-19-related stress reported on here refers to questions answered by mothers specifically about the impact that the COVID-19 pandemic has had on their lives and their overall stress level related to the COVID-19 pandemic (not necessarily the virus). This is a separate measure from general maternal stress, which was also analyzed in the paper discussed. We changed the wording in line 381 to “COVID-19 pandemic-related stress” to clarify that the stress measure discussed is related to pandemic circumstances, not the virus itself.
Line 393-396: The current data? Dose this sentence reflected the previous section (line 371-391)? If so, may I ask the authors how these previous diagnosed “developmental delay” due to maternal exposed COVID-19 become “normal”? Can you add extra comments to the good news? (due to a good response to the rehabilitation, or the inflammation effects automatically attenuation, or the inflammation may not be persist so long as to affect their on-going neurological development?)
While it is correct that one study (87) found differences in ICD-10 diagnoses, we cite several methodological flaws in lines 400-414 that question the validity of this finding. These flaws, combined with the four studies to date that have shown no differences in developmental outcomes between SARS-CoV-2 exposed and unexposed infants (82-84, 86), altogether suggest that there is little evidence to indicate that infants exposed to SARS-CoV-2 in utero are at risk for developmental delay. We clarified this statement in lines 415-424: “Considering these methodological concerns along with the four studies discussed above that have demonstrated no differences in developmental outcomes between in utero SARS-CoV-2 exposed and unexposed infants, the currently available data on long-term outcomes and neurodevelopment in infants exposed to SARS-CoV-2 in utero overall suggests that these infants do not seem to be at any significantly increased risk of neurodevelopmental delay.”
Round 2
Reviewer 1 Report
Dear Authors,
Thank you for submiting the revised version of the manuscript. I have read the revised version of the manuscript. The authors have replied to the comment satisfactorily and made revision as suggested. I think the manuscript is now acceptable for publication.
Author Response
Thank you
Reviewer 2 Report
thanks to the authors for the revised version.
no further comments.
Author Response
Thank you.
Reviewer 3 Report
Accepted
Author Response
Thank you.